# Carbon-anchoring synthesis of Pt₁Ni₁@Pt/C core-shell catalysts for stable oxygen reduction reaction

Jialin Cui[1,4], Di Zhang[2,4], Zhongliang Liu ®[1,4], Congcong Li[1], Tingting Zhang[1], Shixin Yin[1], Yiting Song[1], Hao Li ®[2] ✉, Huihui Li ®[1] ✉ & Chunzhong Li ®[1,3] ✉

Proton-exchange-membrane fuel cells demand highly efficient catalysts for the oxygen reduction reaction, and core-shell structures are known for maximizing precious metal utilization. Here, we reported a controllable "carbon defect anchoring" strategy to prepare Pt₁Ni₁@Pt/C core-shell nanoparticles with an average size of ~2.6 nm on an in-situ transformed defective carbon support. The strong Pt–C interaction effectively inhibits nanoparticle migration or aggregation, even after undergoing stability tests over 70,000 potential cycles, resulting in only 1.6% degradation. The stable Pt₁Ni₁@Pt/C catalysts have high oxygen reduction reaction mass activity and specific activity that reach $1.424 \pm 0.019$ A/mg$_{Pt}$ and $1.554 \pm 0.027$ mA/cm$_{Pt}^2$ at 0.9 V, respectively, attributed to the optimal compressive strain. The experimental results are generally consistent with the theoretical predictions made by our comprehensive microkinetic model which incorporates essential kinetics and thermodynamics of oxygen reduction reaction. The consistent results obtained in our study provide compelling evidence for the high accuracy and reliability of our model. This work highlights the synergy between theory-guided catalyst design and appropriate synthetic methodologies to translate the theory into practice, offering valuable insights for future catalyst development.

Electrocatalytic energy conversion processes play a pivotal role in the development of sustainable technologies, as they offer a pathway towards a renewable future[1–3]. A comprehensive understanding and atomically precise control in the design of catalysts are essential for improving the efficiency of electrochemical steps and related coupling processes[1,4,5]. The utilization of platinum (Pt) as a catalyst for the oxygen reduction reaction (ORR) in proton-exchange membrane fuel cells (PEMFCs) remains challenging because of sluggish kinetics and insufficient activity. One particularly effective solution is to finely tune the surface strain, thus modulating the surface electronic structure and ultimately enhancing catalytic performance[6–8].

The surface strain is determined by the compressed or expanded arrangement of surface atoms, such as core-shell structures with Pt-rich skin. Pt–Ni is one of the most promising bimetallic systems, as proven by some established experiments[9,10]. However, quantitative analysis on the possible active phases and sites was rarely discussed, while most previous theoretical research employed a classic "over-potential model" under a standard hydrogen electrode (SHE) scale by dismissing the experimental fact that ORR performance is generally pH-dependent even at a reversible hydrogen electrode (RHE) scale[11,12]. Meanwhile, the experimental overpotential of ORR is often ill-defined, making the direct benchmarking between experiment and theory almost impossible. Therefore, the development of a unified model that

[1]Key Laboratory for Ultrafine Materials of Ministry of Education, School of Chemical Engineering, East China University of Science and Technology, Shanghai, China. [2]Advanced Institute for Materials Research (WPI-AIMR), Tohoku University, Sendai, Japan. [3]Shanghai Engineering Research Center of Hierarchical Nanomaterials, School of Materials Science and Engineering, East China University of Science and Technology, Shanghai, China. [4]These authors contributed equally: Jialin Cui, Di Zhang, Zhongliang Liu. ✉e-mail: li.hao.b8@tohoku.ac.jp; huihuili@ecust.edu.cn; czli@ecust.edu.cn

considers realistic electrochemical conditions is crucial in accurately predicting the optimal structure of Pt–Ni for achieving the highest ORR performance. Undoubtedly, the challenge of accurately synthesizing pre-designed structures that closely resemble the theoretical prediction poses another significant obstacle in practical implementation.

Herein, we report on a quantitative microkinetic model that considers essential kinetics and thermodynamics of ORR to drive the development of a high-performance Pt–Ni catalyst. According to the simulation results, we developed highly active and stable $Pt_xNi_y$@Pt/C ($x$:$y$ = 1 or 3) core-shell nanoparticles (NPs) catalysts for ORR by a controllable "carbon defect anchoring" strategy to promote the theory into practice. The experimental results are highly consistent with the theoretically predicted activity trends and simulated polarization curves, in which the $Pt_1Ni_1$@Pt/C possesses the highest mass activity and specific activity of $1.424 \pm 0.019$ A/mg$_{Pt}$ and $1.554 \pm 0.027$ mA/cm$_{Pt}^2$, respectively, providing strong evidence for the high accuracy of this model. More importantly, the $Pt_1Ni_1$@Pt/C exhibits an impressive ORR durability with negligible degradation in activity (only 1.6%) over 70,000 potential cycles, without any discernible migration or aggregation of the $Pt_1Ni_1$@Pt/C NPs. The ultra-small $Pt_1Ni_1$@Pt/C NPs ($2.6 \pm 0.6$ nm) firmly anchored on defective carbon substrate through the formation of Pt–C bonds, while the lattice compression strain occurs on the Pt-rich shell, contributing to the enhanced intrinsic activity and stability.

## Results

### Theoretical simulations of target catalyst structures

Firstly, we evaluated the ORR performance of $Pt_xNi_y$@Pt(111) core-shell structures ($x$:$y$ = 1 or 3) and Pt(111) based on a quantitative microkinetic model by considering essential kinetics and thermodynamics of ORR, linear scaling relations between ORR intermediates, pH-field coupled simulations, the potential of zero-charge (PZC), and electrochemical potentials. Details of the computational methods are shown in the Methods. Figure 1a shows a volcano-shaped model as a function of hydroxide (HO*) binding free energy with consideration of the pH effects derived from electric field simulations[11]. The left- and right legs of the volcano are rate-limited by the HO* removal and HOO* formation, respectively. It can be seen that Pt(111) is close to the theoretical maximum of ORR, especially under a low-pH condition. However, pure Pt(111) is still at the left leg of the volcano, with the rate-determining step (RDS) of HO* + H + e⁻ → H₂O. This model aligns with experimentally observed pH-dependent ORR activity and mechanisms on metal surfaces[11,13,14].

We further evaluated the performance of $Pt_xNi_y$@Pt(111) core-shell structures based on this model due to its proven high accuracy. It can be clearly seen that both $Pt_3Ni_1$@Pt(111) and $Pt_1Ni_1$@Pt(111) show high ORR performance, where $Pt_1Ni_1$@Pt(111) is located at the theoretical maximum (i.e., the volcano peak) regardless of the pH (Fig. 1a). Simulating the polarization curves of the three catalysts with consideration of the coverage effects (Fig. 1b), it can be seen that

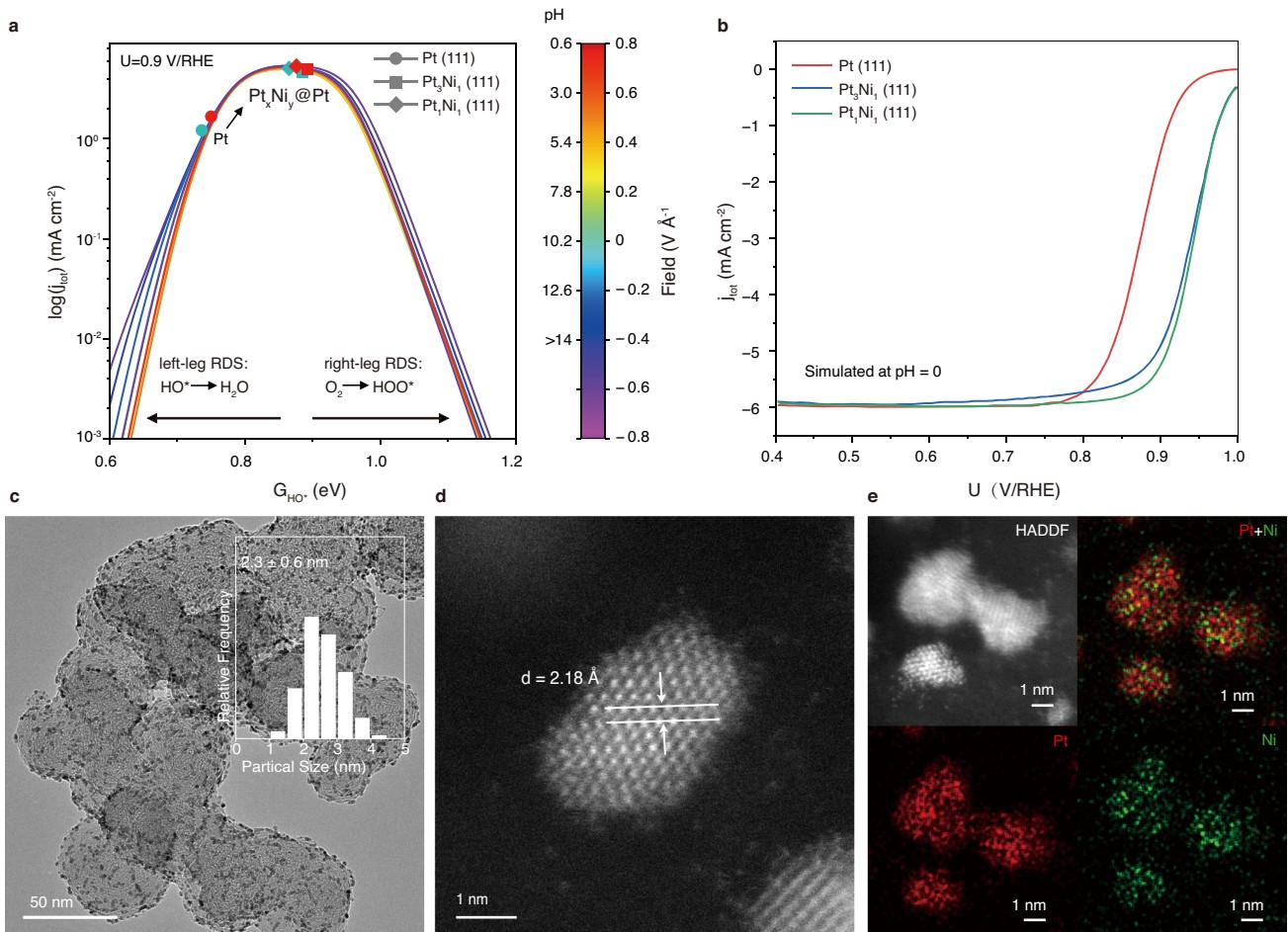

**Fig. 1 | Simulated ORR activities of Pt(111) and $Pt_xNi_y$@Pt(111) based on pH-dependent microkinetic modeling and validation of synthesized $Pt_xNi_y$/C. a** pH-dependent ORR volcano activity model as a function of HO* binding free energy, where a lower electric field corresponds to a lower pH environment, colors of symbol correspond to pH-dependent activity. **b** ORR polarization curves simulated at pH = 0. RDS: rate-determining step. **c** TEM image of $Pt_1Ni_1$/C NPs, inset represents the particle size distribution. **d** Aberration-corrected HAADF-STEM image of $Pt_1Ni_1$/C NPs. **e** HADDF-STEM image of $Pt_1Ni_1$/C NPs and their corresponding elemental mappings. Red Pt, Green Ni.

$Pt_xNi_y@Pt(111)$ core-shell structures significantly outperform Pt(111). The Supplementary Information also includes a simulated volcano plot for the $2e^-$ ORR at 0.6 V vs RHE (Supplementary Fig. 1), which indicates that these Pt/Ni systems are not favorable for the 2-electron reaction and tend to prefer the 4-electron reaction. Because a Ni-rich structure is not stable under acidic ORR conditions, we do not analyze a higher-Ni-ratio Pt−Ni structure in our study[15]. Considering that Pt−Ni is a lower-cost bimetallic compared to pure Pt and can be stable if a pure Pt-shell is formed, these catalysts may show potential for ORR under harsh conditions, such as acidic environments.

### Defect-driven nanostructuring of the catalysts

According to theoretical calculations, the problem and difficulty of the pre-designed catalysts require the feature of homogeneity, surfactant-free, core-shell structure, and high stability. The synthesis method based on high-temperature reduction is effective, but usually leads to non-uniform NPs. It poses a significant challenge on how to utilize the interaction between substrate and metals to restrict nucleation and thus the size of the NPs. As we know, the carbon substrates are important to the size and dispersion of NPs for catalysis applications. However, these NPs may undergo dynamic aggregation into larger sizes due to thermodynamic or kinetic factors. Here, we report a way to prevent the migration and aggregation of the NPs by utilizing defective carbon as anchoring sites for ultra-small $Pt_xNi_y$ NPs (~2.6 nm) through the formation of robust Pt−C bonds.

We developed a fast (within a few minutes), high-intensity sono-chemical synthesis method based on rapid heating rates and thus localized high temperature to deposit ultra-small $Pt_xNi_y$ alloy NPs onto the carbon substrate. The acoustic cavitation from sonochemical treatment causes the formation of bubbles in the reaction system. These bubbles subsequently undergo growth and accumulate energy, then suddenly implode with a transient energy release. The local temperature and pressure can reach ~5000 K with a heating rate > $10^{10}$ K s$^{-1}$ and ~1000 bar, respectively[16]. The production of transient high temperature in this method is crucial for the formation of ultra-small $Pt_xNi_y$ alloy NPs and the simultaneous phase transition of amorphous carbon to highly defective graphite[17–19]. The newly formed defective graphite at the carbon surface provides numerous dangling carbon atoms as ideal nucleation sites, facilitating the capture and interaction with Pt atoms. This exerts a significant influence on the structural and catalysis stability of the catalysts.

The detailed synthetic methods are discussed in Methods. As shown in Fig. 1c and Supplementary Fig. 2, all the synthesized Pt/C (syn-Pt/C), $Pt_3Ni_1$/C, and $Pt_1Ni_1$/C NPs homogeneously deposit on carbon black with a size smaller than 3 nm by transmission electron microscopy (TEM). The atomic resolution structure of $Pt_1Ni_1$/C was unveiled through the aberration-corrected high-angle annular dark-field scanning transmission electron microscopy (HADDF-STEM), which shows a characteristic lattice fringe assigned to the Pt (111) facet (Fig. 1d). The atomic ratio of $Pt_xNi_y$ alloy NPs can be freely changed by varying the mole ratio of metal precursors. The composition of both $Pt_3Ni_1$/C and $Pt_1Ni_1$/C were determined by inductively coupled plasma optical emission spectroscopy (ICP-OES) and energy dispersive spectroscopy (EDS), and the results match well as shown in Supplementary Fig. 3 and Supplementary Table 1. The elemental mapping in Fig. 1e reveals a uniform distribution of Pt and Ni throughout the randomly analyzed region.

A combination of advanced techniques, including X-ray photoelectron spectroscopy (XPS), Raman spectroscopy, electron paramagnetic resonance (EPR) spectroscopy, X-ray adsorption spectrum (XAS), TEM, and X-ray diffraction (XRD), were employed to prove the phase transition from amorphous carbon to highly defective graphite. The $sp^3$-hybridized carbon deconvoluted from C 1$s$ XPS spectra can serve as an indicator for the existence of defective graphite[19,20]. The synthesized carbon-supported catalysts, as demonstrated in Supplementary Figs. 4 and 5, exhibit a higher $sp^3/sp^2$ hybridized carbon ratio compared to the commercial Pt/C (com-Pt/C). This observation suggests that our method effectively yields an increased number of defective carbon anchoring sites. To prove the above point, we further employed Raman spectroscopy to compare the intensity ratio of $D$-band to $G$-band ($I_D/I_G$) of the following samples: fresh carbon black (F-C), ultrasound-assisted treated blank carbon black (US-C), syn-Pt/C, and $Pt_1Ni_1$/C[19,20]. The Raman spectra in Fig. 2a reveal the presence of four distinct peaks, with an $I_D/I_G$ ratio of 1.09 for F-C, which is lower than that observed in all samples after sonochemical treatment. Moreover, the EPR spectroscopy (Fig. 2b), which is highly sensitive to unpaired electrons in coordination with unsaturated structures, exhibits a more intense peak in US-C, demonstrating more carbon defects in the US-C compared to F-C[21,22]. These observations suggest that the high-intensity sonochemical synthesis method effectively promotes the formation of defective carbon anchoring sites.

The C K-edge X-ray absorption near edge structures (XANES) was employed to further elucidate the local structural changes surrounding carbon atoms. Three peaks are identified in Fig. 2c, i.e., $\pi^*$ resonance at ~285.5 eV, $\sigma^*$ resonance at ~293 eV, and the peak at ~288.5 eV. The first peak is associated with the out-of-plane bonds in the $sp^2$ bonding configuration[23], while the negative energy shift and higher intensity observed in the US-C suggest the formation of a disordered carbon lattice and an increased defect density[22]. The $\sigma^*$ resonance corresponds to the in-plane bonds within the hexagonal graphene rings, which are highly sensitive to any structural changes occurring within the same plane[24]. The intensified intensity of this peak in the US-C indicates an increased distortion in the carbon lattice structure. Furthermore, the peak at ~288.5 eV is ascribed to the structural changes due to increased $sp^3$ configuration[25,26]. The TEM analysis helps to illustrate the structural transformation of the carbon framework with twisted and wrinkled graphitic layers at the surface. The arc-shaped stripes observed in F-C correspond to the graphitic layer edges, exhibiting a well-ordered arrangement as indicated in Fig. 2d. In contrast, US-C exhibits numerous twisted and wrinkled structures, indicative of a higher degree of disorder in its carbon structure (Fig. 2e). This graphitic layer structure transformation was also evidenced by XRD patterns in Supplementary Fig. 6, showing a more intensive peak of graphite (002) facet at $2\theta \approx 25°$. These structures are very similar to previously reported work, wherein carbon undergoes a phase transition into turbostratic graphite structures via the Joule heating method[17]. Notably, the synthesis method employed in our study, along with the Joule heating technique, both demonstrate remarkably rapid heating rates and exert an influence on the phase transition of carbon.

Subsequently, precise regulation of the formation of a core-shell structure was achieved through an electrochemical dealloying technique via cyclic voltammetry (CV) scanning to gradually remove Ni atoms from the $Pt_xNi_y$/C alloy NPs, allowing the rearrangement of Pt atoms on the surface of the NPs. This resulted in a simultaneous enhancement of the electrochemically active surface area (ECSA), as illustrated in Fig. 2f and Supplementary Fig. 7. This was further ascertained by electron microscopy coupled with elemental distribution analysis. The EDS line scan analysis across an NP confirms the formation of a well-defined Pt-rich skin structure, suggesting the successful construction of target $Pt_1Ni_1$@Pt/C core-shell structures (Fig. 2g, h). To note, the Pt-to-Ni ratio across the entire NPs increases from ~54.6:45.4 to ~62.3:37.7 after electrochemical dealloying (Supplementary Fig. 8). However, the Pt-to-Ni ratio in the $Pt_1Ni_1$ core remains unchanged, as supported by the calculations and discussion in the Supplementary Notes. Moreover, the crystal structure before and after electrochemical dealloying was examined by XRD, showing a consistent diffraction peak between the face-centered cube (fcc) Pt(111) peak and fcc Ni(111) peak (Supplementary Fig. 9).

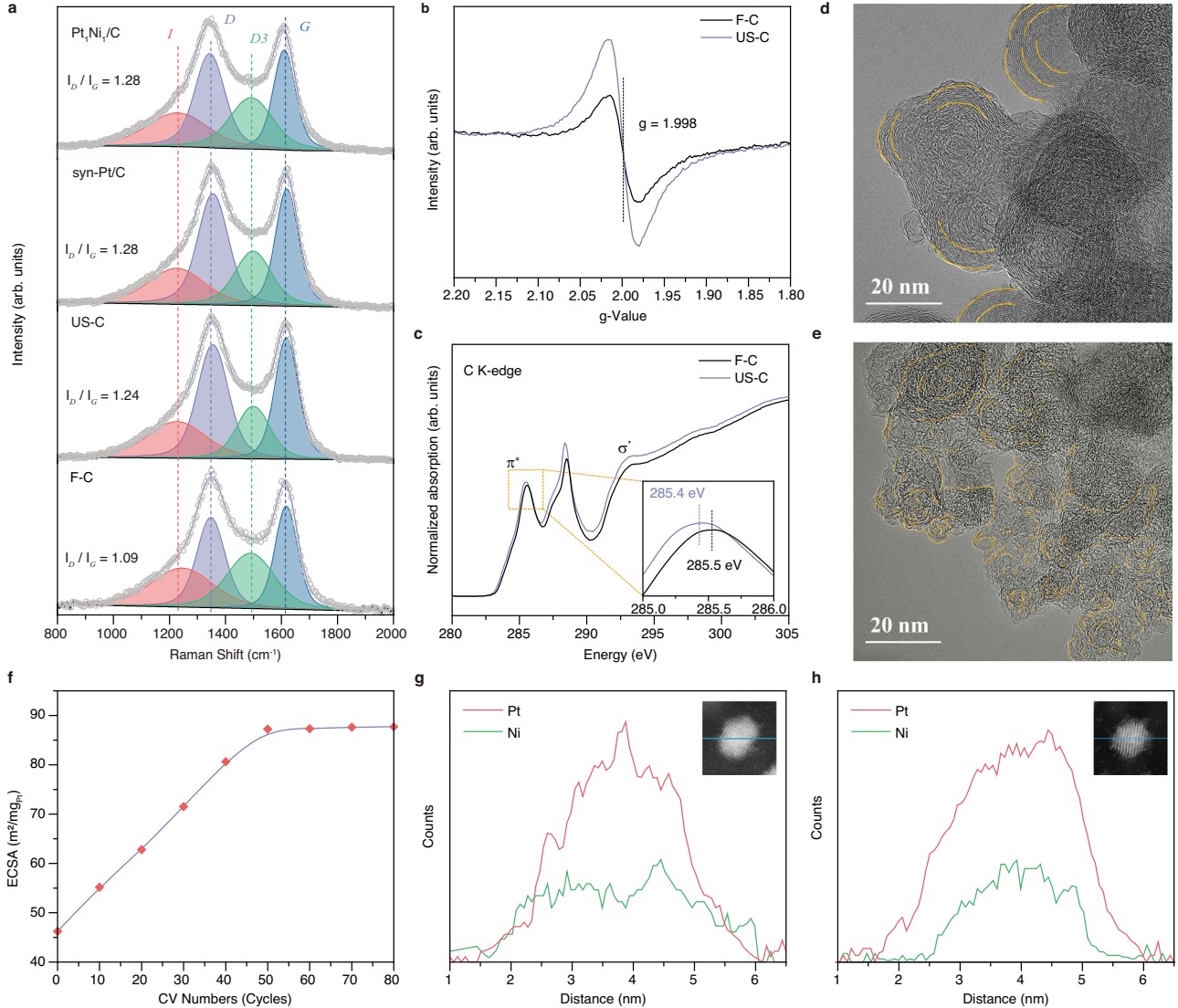

**Fig. 2 | Phase transition of carbon substrate and formation of core-shell structures. a** Raman spectra of F-C, US-C, syn-Pt/C, and Pt₁Ni₁/C, the corresponding $I_D/I_G$ ratio are marked beside, red: *I*-band, purple: *D*-band, green: *D3*-band, blue: *G*-band. **b** EPR spectra of F-C and US-C. **c** XANES spectra at the C K-edge of F-C and US-C, inset is a magnified view of the selected region. TEM images of (**d**) F-C and **e** US-C. **f** Relationship between ECSA and CV cycles of Pt₁Ni₁/C during electrochemical dealloying. EDS line scan analysis of (**g**) as-prepared Pt₁Ni₁/C NPs and **h** electrochemical-dealloying Pt₁Ni₁@Pt/C, inset is the HADDF-STEM image of the analyzed NP.

## ORR performance of Pt₁Ni₁@Pt/C core-shell catalysts

Structural studies were performed to characterize the Pt₁Ni₁@Pt/C and pure syn-Pt/C catalysts, demonstrating that the size and structure were well maintained after the electrochemical dealloying process as depicted in Fig. 3a. The HADDF-STEM analysis reveals that the lattice spacing of Pt(111) in Pt₁Ni₁@Pt/C is measured to be 2.18 Å, which is shorter than that of syn-Pt/C (2.28 Å). Therefore, the presence of lattice shrinkage in the obtained core-shell structure confirms the existence of compressive strain (Fig. 3a, b). The compressive strain exhibited a strong correlation with the enhanced ORR activity, which resulted in the reduction of adsorption energetics for HO* at Pt sites based on the d-band model[9]. So, the compression strain may be the primary driving force propelling the Pt₁Ni₁@Pt(111) towards the peak of the volcano activity model, as depicted in Fig. 1a.

The ORR electrocatalytic performance was evaluated on a rotating disk electrode (RDE) in 0.1 M HClO₄ solution. Four samples were tested: Pt₁Ni₁@Pt/C, Pt₃Ni₁@Pt/C syn-Pt/C, and com-Pt/C catalysts. The synthesized catalysts all exhibit similar particle size (Supplementary Fig. 2), thereby excluding the influence of size effect on ORR

performance. Solution resistance of each catalyst was measured by electrochemical impedance spectroscopy (EIS) as presented in Supplementary Fig. 10a. Experimentally, as demonstrated by the *i*R-corrected ORR polarization curves in Fig. 3c and non-*i*R corrected curves in Supplementary Fig. 10b, Pt₁Ni₁@Pt/C catalyst is found to be considerably more active than Pt₃Ni₁@Pt/C and syn-Pt/C catalysts, thereby highlighting the high accuracy of our quantitative micro-kinetic model. The theoretically optimal Pt₁Ni₁@Pt/C and Pt₃Ni₁@Pt/C catalysts exhibit high half-wave potentials of $0.927 \pm 0.001$ V and $0.908 \pm 0.002$ V vs RHE, respectively, surpassing that of the syn-Pt/C ($0.872 \pm 0.005$ V vs RHE) and the com-Pt/C ($0.858 \pm 0.005$ V vs RHE) catalysts as detailed in Supplementary Table 2. The corresponding Tafel plots and slopes are presented in Fig. 3d and Supplementary Table 2, wherein the Pt₁Ni₁@Pt/C displays lower Tafel slopes owing to their enhanced kinetics resulting from appropriate adsorption energetics of HO* compared to pure Pt.

To assess the intrinsic ORR activity, the ECSA of all the catalysts was quantified based on the charge collected in the hydrogen adsorption/desorption region in CV curves (Fig. 3e and Supplementary

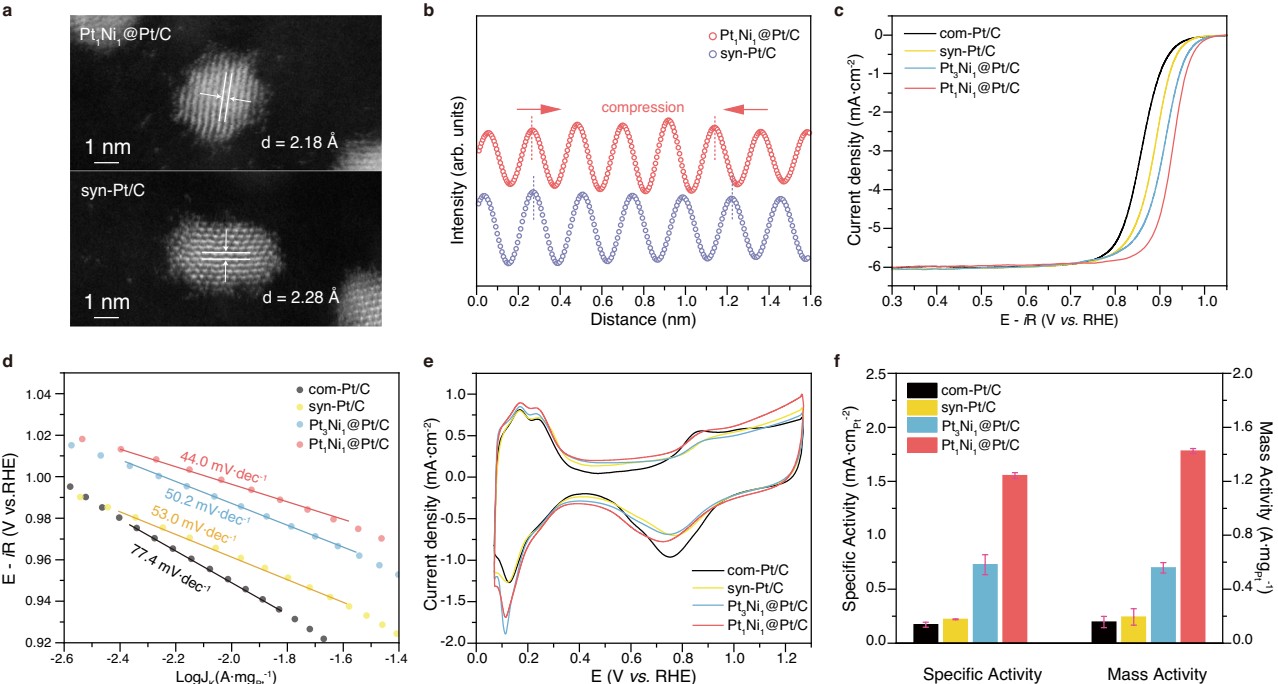

**Fig. 3 | Experimental ORR activities of Pt$_1$Ni$_1$@Pt/C, Pt$_3$Ni$_1$@Pt/C, syn-Pt/C, and com-Pt/C. a** Aberration-corrected HADDT-STEM image of Pt$_1$Ni$_1$@Pt/C and syn-Pt/C NPs. **b** HADDF intensity profile of Pt$_1$Ni$_1$@Pt/C and syn-Pt/C. **c** *i*R corrected ORR polarization curves (**d**) Tafel plots, **e** CV curves, and **f** specific and mass activities of different catalysts measured in 0.1 M HClO$_4$. The error bars indicate the standard deviation, derived from at least three independent measurements. *i*R correction was conducted at 100% manually.

Fig. 11[27]. The Pt$_1$Ni$_1$@Pt/C and Pt$_3$Ni$_1$@Pt/C catalysts display a little higher and similar ECSA of $87.2 \pm 0.3$ m$^2$/g$_{Pt}$ and $79.0 \pm 1.4$ m$^2$/g$_{Pt}$, respectively, which are comparable with that of the pure Pt catalysts: syn-Pt/C ($73.9 \pm 3.1$ m$^2$/g$_{Pt}$), and com-Pt/C ($74.8 \pm 0.9$ m$^2$/g$_{Pt}$). Undoubtedly, the increase in ECSA is attributed to the successful formation of a Pt-rich shell over the surface of Pt$_x$Ni$_y$@Pt/C catalysts (Fig. 2f). The kinetic currents were normalized with respect to both ECSA and the metal Pt loading amount to facilitate a comparative analysis of catalyst activity. As shown in Fig. 3f and Supplementary Table 2, the specific activity ($1.554 \pm 0.027$ mA/cm$_{Pt}^2$) and mass activity ($1.424 \pm 0.019$ A/mg$_{Pt}$) of Pt$_1$Ni$_1$@Pt/C are nearly ~8.0 and ~8.1 times higher than that of syn-Pt/C ($0.172 \pm 0.022$ mA/cm$_{Pt}^2$ and $0.157 \pm 0.042$ A/mg$_{Pt}$), respectively. In addition, to further verify the accuracy of the pH-dependent quantitative microkinetic model, the ORR activity in 0.1 M KOH electrolyte was also evaluated, and the obtained results exhibit consistency with those predicted by theoretical calculations (Supplementary Figs. 10c and 12 and Supplementary Table 3). The compression strain may be responsible for the enhanced intrinsic activity of Pt$_1$Ni$_1$@Pt/C core-shell structures via enhanced oxidative desorption ability of HO*[9,28,29].

The assessment of ORR durability was conducted by accelerated durability test (ADT) between 0.6 V and 1.0 V vs RHE at 100 mV/s in O$_2$-saturated 0.1 M HClO$_4$ solution. The CVs and polarization curves were measured and the ECSA, mass activity, and specific activity were calculated after every 10,000 potential cycles. Figure 4a, b and Supplementary Fig. 13a display no obvious deterioration in ECSA and ORR activity of Pt$_1$Ni$_1$@Pt/C after 70,000 potential cycles. In contrast, the ECSA and activity of com-Pt/C rapidly decay after only 30,000 potential cycles (Fig. 4c, d and Supplementary Fig. 13b). The mass activity of the Pt$_1$Ni$_1$@Pt/C exhibited a minimal decrease of only ~1.6% after 70,000 cycles, surpassing the performance degradation observed in com-Pt/C (38.4% reduction after 30,000 potential cycles) and outperforming recently reported Pt-based ORR catalysts, as illustrated in Fig. 4e and Supplementary Table 4. The stability of com-Pt/C was found to be inferior to that of Pt$_1$Ni$_1$@Pt/C, which may be due to

the occurrence of aggregation and Ostwald ripening processes in com-Pt/C catalysts (Supplementary Fig. 14)[30,31]. Despite they have similar spherical NPs, the successful formation of Pt–C bonds between Pt$_1$Ni$_1$@Pt/C and defective carbon can effectively prevent the movement and detachment of the NPs on the carbon substrate, contributing to improved stability. TEM studies conducted on Pt$_1$Ni$_1$@Pt/C and syn-Pt/C catalysts before and after ADT tests demonstrate negligible alterations in the morphology and size (Supplementary Figs. 15 and 16). Moreover, the formed Pt-rich shell also serves as a protective layer to suppress the dissolution of the Pt$_1$Ni$_1$ core.

To track the composition evolution of the catalysts after the durability tests, we examined the composition variation of the catalysts after 10,000, 30,000, 50,000, and 70,000 ADT cycles, respectively (Supplementary Fig. 17). The ongoing etching of Ni has resulted in significant compositional alterations in atomic ratio of Ni from 37.7% to 8.6% during ADT tests. However, the changes in composition do not adversely affect the activity after 70,000 ADT cycles. So, we concluded that the robust structural stability of the synthesized Pt$_1$Ni$_1$@Pt/C catalysts is crucial for the stable ORR, which may compensate for the leaching of non-precious metal components.

We speculated that such enhanced stability is attributed to the strong interaction of the defective graphic layer with Pt$_1$Ni$_1$@Pt/C NPs through the formation of Pt–C bonds. The presence of a defective graphene layer, resulting from intense sonochemical treatment, offers an abundance of dangling carbon atoms that effectively capture Pt atoms. To validate our speculation, we firstly examine the Pt oxidation states in the synthesized Pt$_x$Ni$_y$/C by calculating integral area of white line peak in Pt L$_3$-edge XANES spectra as depicted in Fig. 5a and Supplementary Fig. 18. Comparing to standard reference Pt and PtO$_2$, the Pt$_1$Ni$_1$/C and Pt$_3$Ni$_1$/C exhibit slightly higher valence state of +0.74 and +0.23, respectively, implying an electron transfer between Pt atoms and carbon substrate or Ni atoms. To provide a comprehensive explanation of the above findings, further analysis was conducted on the FT-EXAFS spectrum at the Pt L$_3$-edge of Pt$_1$Ni$_1$/C. The results revealed that the first peak can be attributed to the Pt–C bond, with

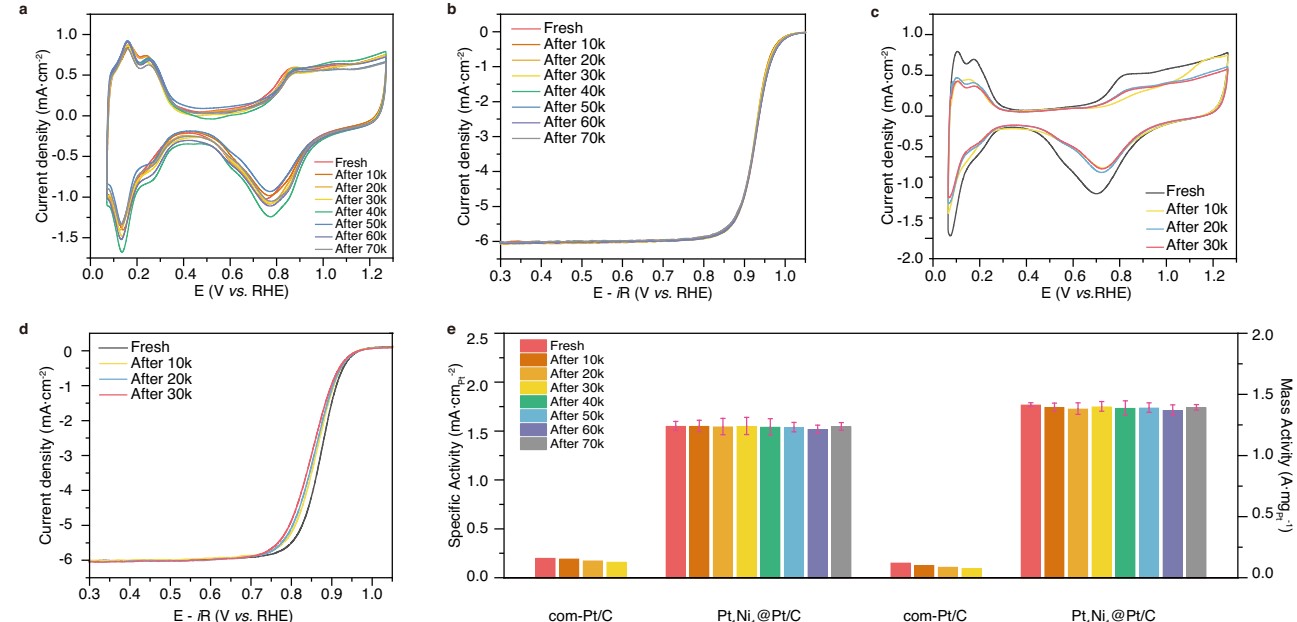

**Fig. 4 | ORR durability of Pt$_1$Ni$_1$@Pt/C and com-Pt/C. a** CV curves and **b** *i*R corrected ORR polarization curves of Pt$_1$Ni$_1$@Pt/C after every 10 k potential cycles in 0.1 M HClO$_4$. **c** CV curves and **d** *i*R corrected ORR polarization curves of com-Pt/C after every 10 k potential cycles in 0.1 M HClO$_4$. **e** MA and SA comparison between Pt$_1$Ni$_1$@Pt/C and com-Pt/C during ADT. The error bars indicate the standard deviation, derived from three independent ADT measurements. *i*R correction was conducted at 100% manually.

reference to com-Pt/C and standard Pt foil (Fig. 5b). Moreover, the Pt−C interaction is also confirmed by Pt 4*f* XPS spectra as shown in Fig. 5c, the Pt 4$f_{7/2}$ peak of syn-Pt/C is positively shifted to 72.2 eV compared to Pt$^0$ at ~71.1 eV[32]. The comprehensive X-ray spectroscopy characterizations clearly elucidate the robust interaction between Pt and the carbon substrate, thereby facilitating the notably stable ORR performance in acidic media.

We further implement ab initio molecular dynamics (AIMD) simulations to investigate the indispensable function of carbon defects in bonding Pt atoms (computational details can be found in Supplementary Methods). As illustrated in Fig. 5d, e, the defected Pt-graphene system is prone to form Pt−C bonds while the defect-free system remains unchanged after 1000-fs AIMD simulations. These results suggest that high-intensity sonochemical treatment plays an important role in generating carbon defects and promoting bonding between metal NPs and defect sites. However, it should be noted that the exact nature of these interactions may differ under practical catalytic conditions, where additional factors such as electrolyte effects and prolonged operational stresses may influence the stability and effectiveness of these bonds. Furthermore, considering the carbon phase transition after high-intensity sonochemical treatment, the twisted and wrinkled graphitic layers at the carbon black surface may provide enough spacing for metal atoms intercalating under the transient heating conditions[17]. This phenomenon is also observed in the TEM image of Pt$_1$Ni$_1$/C NPs, resembling those of carbon-supported metallic NPs synthesized by the Joule heating method[17]. Three Pt$_1$Ni$_1$ NPs positioned at the edges of the carbon substrate are highlighted in Fig. 5f, displaying partial embedding within the carbon matrix. This observation further reinforces the presence of strong bonding between Pt and carbon support, ensuring the prolonged durability of the final Pt$_1$Ni$_1$@Pt/C in ORR.

## Discussion
This work presents a comprehensive microkinetic model and provides deep insights into the higher ORR performance of Pt$_x$Ni$_y$@Pt (111) catalysts under both the acidic and alkaline condition, driving us to design and synthesize highly active core-shell catalysts. We developed a controllable sonochemical synthesis method followed by electrochemical dealloying to prepare Pt$_x$Ni$_y$@Pt/C core-shell catalysts on a defective carbon substrate. In this method, we elucidate the simultaneous processes of defect generation, amorphous carbon graphitization, and anchoring of ultra-small NPs by forming robust Pt−C bonds during ultrafast exposure to the localized high temperature.

The final Pt$_1$Ni$_1$@Pt/C catalyst demonstrated a notable improvement in both mass and specific activity, achieving $1.424 \pm 0.019$ A/mg$_{Pt}$ and $1.554 \pm 0.027$ mA/cm$_{Pt}^2$, respectively. This can be attributed to the compressive strain induced by the lattice shrinkage between the Pt-rich shell and alloy core, which weakens the adsorption strength of oxygen-intermediates with Pt skin and thus contributes to enhanced intrinsic activity. The strong interaction between Pt and C in the Pt$_1$Ni$_1$@Pt/C catalysts is primarily responsible for the significant enhancement of ORR durability, exhibiting minimal degradation even after 70,000 potential cycles. This work integrates the theoretical guidance in catalyst design with experimental synthesis, providing valuable insights for the theory-driven rational design of catalysts and highlighting the significance of selecting appropriate synthetic methods. It should also be acknowledged that these models are based on static surface configurations that may not fully represent the dynamic nature of the catalyst surface under operational conditions.

## Methods
### Synthesis of Pt$_x$Ni$_y$/C and Pt/C NPs
In a typical preparation of syn-Pt/C, chloroplatinic acid hexahydrate (H$_2$PtCl$_6$·6H$_2$O, 7.8 mg), Vulcan XC-72 (25 mg), and 30 mL ethylene glycol were added into a vial (volume: 50 mL). The pH was adjusted to 9 by the gradual addition of 1 M NaOH aqueous solution. The above mixture was ultrasonicated for about 1 h in an ultrasonic cleaner to ensure homogenous dispersion. For the high-intensity sonochemical treatment, a cylindrical tip with an 8 mm diameter was employed for 15 min, operating at 60% amplitude. The parameters were set with a power of 750 W and a frequency of 23.76 kHz. After the high-intensity sonochemical treatment, the synthesized products were collected by vacuum suction filtration and washed three times with water and ethanol.

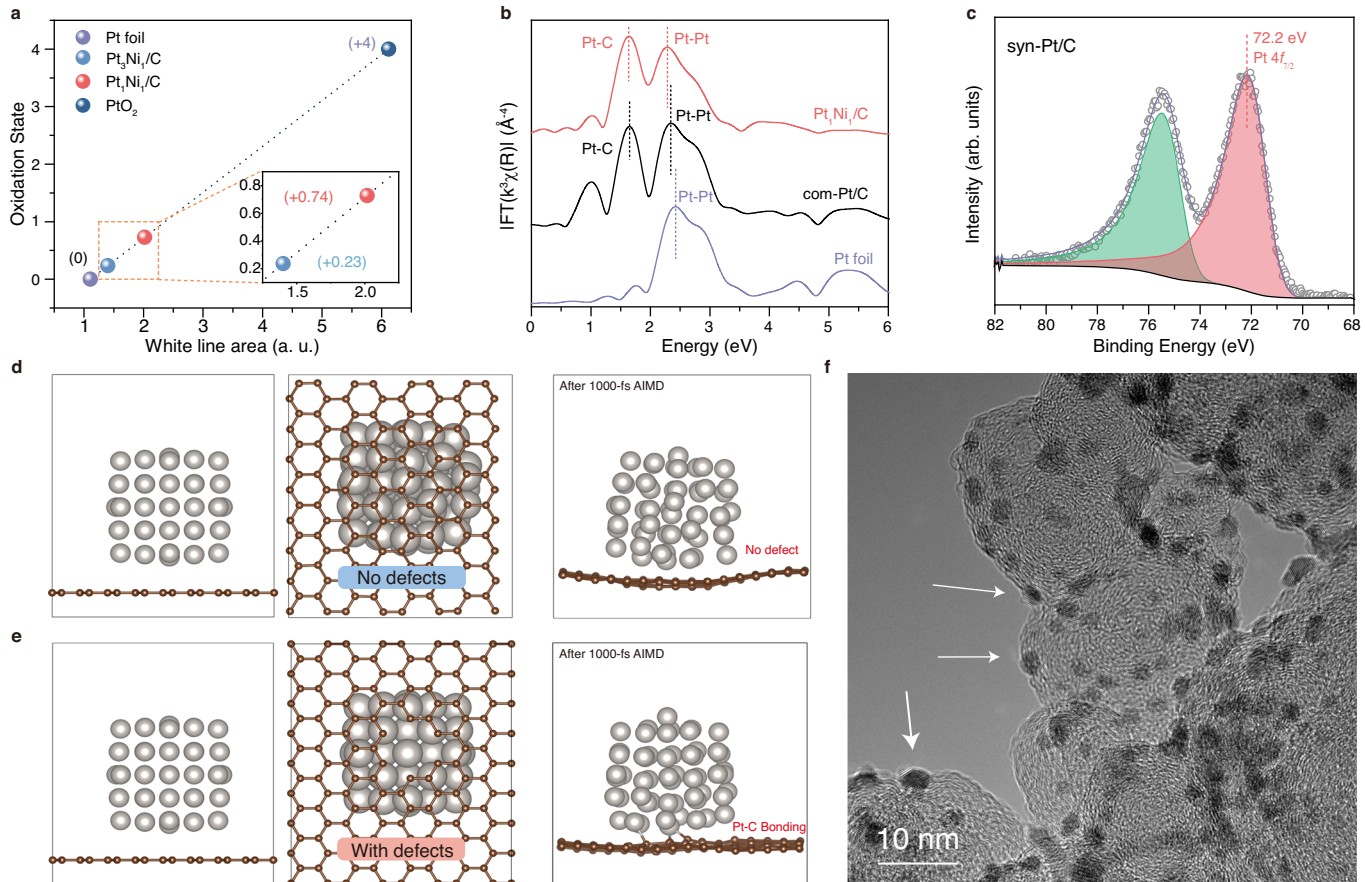

**Fig. 5 | Strong interactions between Pt₁Ni₁ NPs and carbon substrate. a** XANES spectra at the Pt L₃-edge of Pt₁Ni₁/C, Pt₃Ni₁/C, standard Pt foil, and standard PtO₂, inset is a magnified view of selected region. **b** FT-EXAFS spectra at the Pt L₃-edge of Pt₁Ni₁/C, com-Pt/C, and Pt foil. **c** Pt 4*f* XPS spectra of syn-Pt/C. Side view, bottom view of **d**, **e** the defect-free and the defected Pt-graphene systems and their results after 1000-fs AIMD simulations, respectively. **f** TEM image of Pt₁Ni₁/C, the embedded Pt₁Ni₁ NPs are pointed out.

For the synthesis of Pt$_x$Ni$_y$/C, all the conditions were similar to those used for the synthesis of syn-Pt/C, except that nickel (II) nitrate hexahydrate was added in the corresponding proportion.

### Synthesis of Pt$_x$Ni$_y$@Pt/C core-shell structures

The Pt$_x$Ni$_y$@Pt/C core-shell structures were formed through electrochemical dealloying. Firstly, a catalyst ink was prepared by mixing the 3 mg Pt$_x$Ni$_y$/C with 990 μL 2-propanol and 10 μL Nafion solution (5 wt%). The resulting suspension was then subjected to ultrasonication for 30 min using an ultrasonic cleaner. The Pt mass content in the catalyst ink was determined by ICP-OES prior to electrode fabrication. A specific volume of the suspension was then drop-cast onto a glassy carbon rotating disk electrode (RDE) with a 5 mm diameter and a geometric area of 0.196 cm², resulting in Pt loadings of approximately 15 μg/cm² on the working electrode. A Pt foil and an Ag/AgCl (3.5 M KCl) electrode were used as the counter and reference electrodes, respectively. The electrochemical dealloying process was carried out in N₂-saturated 0.1 M HClO₄ electrolyte using CV between −0.2 V and 1.0 V vs Ag/AgCl with a scan rate of 100 mV/s for 50 cycles in a single-compartment, five-neck cell.

### Electrochemical measurements

All the electrochemical measurements were measured on an electrochemical workstation (CHI760e) at room temperature. The setup of the three-electrode system is the same as that used in electrochemical dealloying. Pt loadings were controlled at 15 μg/cm² for all samples. The calibration of the Ag/AgCl (3.5 M KCl) reference electrode was conducted under an H₂ atmosphere, with Pt foil as the working electrode. All potentials were referenced to the RHE and the conversion between the Ag/AgCl reference electrode potential ($E_{Ag/AgCl}$) and RHE potential ($E_{RHE}$) was $E_{RHE} = E_{Ag/AgCl} + 0.270V$.

The 0.1 M HClO₄ electrolyte was freshly prepared before each electrochemical measurement by diluting 70–72% perchloric acid with deionized water.

The ORR activity was measured by linear sweep voltammetry (LSV) in O₂-saturated 0.1 M HClO₄ electrolyte at a rotation speed of 1600 rpm and a potential scan rate of 10 mV/s without *iR* compensation. All currents were corrected by subtracting the background current obtained in N₂-saturated 0.1 M HClO₄. The solution resistance was measured using EIS at 0 V vs Ag/AgCl, with a frequency range of 100 kHz to 1 Hz and an amplitude of 5 mV. All polarization curves were 100% *iR* corrected manually.

An accelerated durability test (ADT) was performed in an O₂-saturated 0.1 M HClO₄ electrolyte with a potential scan rate of 100 mV/s in the potential range of 0.6–1.0 V vs RHE at room temperature.

The electrochemical active surface area (ECSA) was measured in N₂-saturated 0.1 M HClO₄ electrolyte by CV between −0.2 V and 1.0 V vs Ag/AgCl with a scan rate of 50 mV/s. The ECSA was calculated using the following formula:

$$ECSA_{Hupd} = \frac{aECSA}{M_{Pt}} = \frac{Q_H}{CD_{Hupd} \times M_{Pt}}$$

Where $Q_H$ is derived by integrating the H adsorption region from the CV curve over the potential range of 0.05–0.40 V vs RHE. $M_{Pt}$

represents the absolute Pt loading ($mg_{Pt}$). $CD_{Hupd}$ refers to the charge density of Hupd, which is 210 μC/cm² for polycrystalline Pt.

The kinetic current density ($j_k$) is calculated using the following equation:

$$j_k = \frac{j_L \times j}{j_L - j}$$

Where $j$ represents the current density at 0.9 V vs RHE obtained from polarization curves. $J_L$ represents the diffusion-limited current density.

Mass activity (MA) and specific activity (SA) are calculated using the following equations, respectively.

$$MA = \frac{j_k}{m_{Pt}}$$

$$SA = \frac{M_A}{ECSA}$$

Where $m_{Pt}$ represents the Pt mass loading per unit area.

### Computational methods
Spin-polarized density functional theory (DFT) calculations were performed with the VASP code. To deal with the electronic exchange-correlation, a revised Perdew–Burke–Ernzerhof (RPBE) functional with the generalized gradient approximation (GGA) method was employed for all the calculations[33,34]. Core electrons were represented based on the projector-augmented wave (PAW) method[35], while the valence electrons were considered by expanding the Kohn–Sham wave functions by plane-wave basis set[36]. The $k$-point mesh sampling was $2 \times 2 \times 1$ to sample the Brillouin zone using the method by Monkhorst and Pack. The energy cutoff of all calculations was 400 eV. The force convergence was reached after all the forces of each atom fell below 0.05 eV per Å. All the Pt-based surfaces were modeled as four-layer, $4 \times 4$ slab models, with the bottom two layers fixed in bulk positions. To avoid the periodic effects in the $z$-direction, a vacuum of at least 14 Å was modeled. The lattice constants of the models were chosen based on Vegard's law with the experimental values from the database. Stricter criteria (e.g., larger cutoff, larger $k$-points, the thicker slabs) were tested; no significant difference was found in the energetics or optimized structures.

The computational hydrogen electrode method developed by Nørskov et al.[37]. was employed for all the free energy calculations. Zero-point energy and entropic corrections (at 298 K) were included based on the values from previous studies[11,12]. The pH-dependent volcano activity model (i.e., the microkinetic model as a function of pH and HO* binding energy) for ORR was developed based on the method by Kelly et al.[11] using the CatMap package[38]. The quantitative accuracy of this pH-dependent volcano model was proven by the benchmarking analyses between theory and experiments on Pt(111) and Au(100) under various pH conditions. The linear scaling relations between HO* binding energy and the binding energies of all other intermediate species (e.g., O$_2$*, HOO*, and O*) were obtained from previous studies[11]. The dipole moment and polarizability information were acquired from the calculated adsorption energies applied under various electric fields on Pt(111). The PZC value was obtained from previous experiments on metal surfaces[11].

### Data availability
All relevant data generated in this study are provided in the Supplementary Information/Source Data file. Source data are provided in this paper, and computational source data can be accessed at www.digcat.org. Source data are provided with this paper.

### Code availability
All the atomic coordinates of the optimized computational models and dynamics trajectories are available at https://github.com/tohokudizhang/PtNix_ORR[39].

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

## Acknowledgements

This work was supported by the National Natural Science Foundation of China (U22B20143, 21838003, and 21771170), the Shanghai Municipal Science and Technology Major Project, the Shanghai Scientific and Technological Innovation Project (22dz1205900), the Shanghai Rising-Star Program (20QA1402700), JSPS KAKENHI (no. JP23K13703), and the Hirose Foundation. The authors thank the Shanghai Synchrotron Radiation Facility (14W1, SSRF) and the Center for Computational Materials Science, Institute for Materials Research, Tohoku University for the use of MASAMUNE-IMR (project nos. 202312-SCKXX-0203 and 202312-SCKXX-0207) and the Institute for Solid State Physics (ISSP) at the University of Tokyo for the use of their supercomputers. D.Z. acknowledges the National Natural Science Foundation of China (no. 22309109) and KAKENHI Start-Up (no. JP24K23068). D.Z. gratefully acknowledges the support provided by the Shanghai Jiao Tong University Outstanding Doctoral Student Development Fund.

## Author contributions

Conceptualization: J.C., H.L., H.H.L., and C.L. Methodology: J.C. and Z.L. Investigation: J.C., D.Z., Z.L., C.L., T.Z., S.Y., and Y.S. Visualization: J.C., Z.L., and D.Z. Software: D.Z. and H.L. Supervision: H.L., H.H.L., and C.L. Writing—original draft: J.C., Z.L., and D.Z. Writing—review and editing: J.C., Z.L., H.L., H.H.L., and C.L.

## Competing interests

The authors declare no competing interests.
