## [Peer Review File · Nature Communications]

REVIEWER COMMENTS

Reviewer #1 (Remarks to the Author):

In this article, the authors proposed a carbon-anchoring synthesis of Pt₁Ni₁@Pt/C core-shell catalysts, which is optimized based on a quantitative microkinetic model that consider essential kinetics and thermodynamics of ORR. Interestingly, they implemented a fast, room-temperature, high-intensity sonochemical method followed by electrochemical dealloying to precisely synthesize the theoretically optimal core-shell structure. The obtained Pt₁Ni₁@Pt/C core-shell catalyst exhibits enhanced ORR activity, which is in good agreement with theoretical predictions. Impressively, the remarkable durability of Pt₁Ni₁@Pt/C core-shell catalyst is studied by comprehensive characterizations. The elucidation of how carbon defects induced by high-intensity sonication strongly bond metal nanoparticles during the synthesis process is clearly outlined. This study presents new findings and interesting results. Therefore, I believe these results should be of interest to the broad readership of Nature Communications. I am pleased to recommend its publication after addressing several minor revisions as outlined below.

1. In this work, the authors present a volcano-shaped model in Fig. 1a, illustrating the relationship between HO* binding free energy and pH effects resulting from electric field simulation. They also asserted that they have developed a comprehensive model that considers realistic electrochemical conditions, thereby facilitating accurate prediction of the optimal Pt-Ni structure. But, in Fig. 1a and S1, it is difficult to know the specific pH range indicated by "low pH" and "high pH". I suggest providing quantitative values in Fig. 1a for better clarity.

2. According to the results in Fig. 1a, the Pt₁Ni₁@Pt(111) is identified as the theoretically optimal structure for ORR, regardless of pH. However, the authors have only report the activities of Pt_xNi_y@Pt/C in acidic electrolyte, which are consistent with theoretical predictions. It remains unclear what the activities of Pt_xNi_y@Pt/C are in alkaline media and whether they follow the activity trend predicted by theory.

3. The authors claim that the instability of com-Pt/C is associated with movement, aggregation, and Ostwald ripening process. Could this be the reason why commercial catalysts experience degradation after undergoing 10,000 cycles of stability testing? It is necessary to furnish supporting evidence for this claim.

4. Some minor errors: (1) In Fig. 3a, the PtNi@Pt/C should be Pt₁Ni₁@Pt/C? (2) The units of half-wave potential should be supplemented with the designation "vs. RHE". (3) The Supplementary Information contains a reference to AIMD results on line 9 of page 3, which incorrectly cites Fig. 4 in the manuscript instead of Fig. 5.

Reviewer #2 (Remarks to the Author):

In this paper, the authors present a synthesis method for PtNi/C catalysts with high dispersion and uniform size and propose a microkinetic model to predict the performance of these catalysts. The prepared catalysts exhibited very high activity in the oxygen reduction reaction (ORR) and demonstrated significantly superior durability compared to previously reported platinum alloy catalysts for the oxygen reduction reaction. However, the following aspects need to be addressed for the publication of this

paper.

1. The synthesized PtNi/C catalysts underwent a dealloying process before the evaluation of ORR performance. Since dealloying removes nickel from the alloy particles, the actual platinum-to-nickel ratio of the catalysts is likely different. For the proposed microkinetic model to be appropriate, it is necessary to measure the platinum-to-nickel ratio of the dealloyed PtNi/C catalysts.
2. The crystallographic characteristics of the prepared catalysts need to be provided. It is essential to measure and discuss the XRD of the catalysts before and after dealloying.
3. It is known that the durability degradation of platinum alloy catalysts is more influenced by the leaching of non-precious metal components than by the aggregation of alloy particles. However, in this paper, the authors claim that the high durability of the synthesized PtNi/C catalysts is due to the suppression of aggregation by the strong interaction between carbon (defect sites) and platinum alloy particles. This needs to be clarified through a compositional analysis of the catalysts after the durability tests.
4. If the defect sites generated during the catalyst preparation inhibit metal aggregation during durability tests, then the synthesized Pt/C should exhibit similar durability behavior as Pt₁Ni₁/C. Therefore, the authors should conduct durability tests on the synthesized Pt/C catalysts and provide an examination of the morphology before and after the tests.
5. In the durability tests, the authors used a Pt foil as the counter electrode. Using a Pt foil as the counter electrode, especially in durability tests, makes accurate evaluation difficult due to platinum redeposition. Therefore, authors should discuss the result of the durability test conducted with a carbon as counter electrode.
6. Catalysts that exhibit high performance in half cell do not always perform similarly in unit cells. Since the platinum alloy catalysts in this paper are intended for fuel cells, the performance evaluation results of unit cells using the synthesized catalysts should be provided.

Reviewer #3 (Remarks to the Author):

Cui et al. used a quantitative microkinetic model to estimate the oxygen reduction reaction (ORR) performance of Pt_xNi_y@Pt(111) and synthesized the corresponding catalysts using a sonochemical approach. This work lacks novelty, as similar results from simulations and experiments have been reported frequently in the literature. The highlighted importance of Pt-C interactions in enhancing ORR catalyst durability is also validated through fuel cell tests. I recommend rejection of this manuscript.

1. Why the simulated polarization curves for Pt₃Ni(111) and Pt₁Ni₁(111) in Fig.1b are not parallel.
2. There seems to be contamination affecting the durability test of Pt/C (Fig.6d). The observed degradation behavior is unusual and merits further investigation.
3. The authors should include X-ray diffraction (XRD) results for the Pt_xNi_y@Pt(111) catalysts to provide more comprehensive characterization data.
4. The authors attribute the partial embedding of Pt particles within the carbon matrix to strong Pt-C bonds (as indicated in Fig. 5f). This explanation seems inaccurate; partial embedding is often due to overlapping and is commonly observed in commercial Pt/C catalysts.

RESPONSE TO REVIEWERS

Response to Reviewer #1

Reviewer #1

In this article, the authors proposed a carbon-anchoring synthesis of Pt₁Ni₁@Pt/C core-shell catalysts, which is optimized based on a quantitative microkinetic model that consider essential kinetics and thermodynamics of ORR. Interestingly, they implemented a fast, room-temperature, high-intensity sonochemical method followed by electrochemical dealloying to precisely synthesize the theoretically optimal core-shell structure. The obtained Pt₁Ni₁@Pt/C core-shell catalyst exhibits enhanced ORR activity, which is in good agreement with theoretical predictions. Impressively, the remarkable durability of Pt₁Ni₁@Pt/C core-shell catalyst is studied by comprehensive characterizations. The elucidation of how carbon defects induced by high-intensity sonication strongly bond metal nanoparticles during the synthesis process is clearly outlined. This study presents new findings and interesting results. Therefore, I believe these results should be of interest to the broad readership of Nature Communications. I am pleased to recommend its publication after addressing several minor revisions as outlined below.

Response: We sincerely appreciate your positive evaluation and insightful comments on our manuscript and your recommendation for publication in *Nature Communications*. For your convenience, the main revisions are marked in red in the **revised Manuscript** and **Supplementary Information**.

(1) In this work, the authors present a volcano-shaped model in Fig. 1a, illustrating the relationship between HO* binding free energy and pH effects resulting from electric field simulation. They also asserted that they have developed a comprehensive model that considers realistic electrochemical conditions, thereby facilitating accurate prediction of the optimal Pt-Ni structure. But, in Fig. 1a and S1, it is difficult to know the specific pH range indicated by "low pH" and "high pH". I suggest providing quantitative values in Fig. 1a for better clarity.

Response: We appreciate your excellent suggestion. In the updated version, we have modified Fig. 1a in the **revised Manuscript** and Supplementary Fig. 1a in the **revised Supplementary Information** to directly display the pH values corresponding to the electric field in the figures. As shown in **Fig. R1a** and **R1b**, we have now included quantitative pH values to provide better clarity.

Fig. R1. pH-dependent ORR volcano activity model as a function of HO* binding free energy, with the relationship between electric field and pH shown in the color bar. (a) 4 e⁻ ORR volcano (b) 2 e⁻ ORR volcano.

(2) According to the results in Fig. 1a, the Pt₁Ni₁@Pt(111) is identified as the theoretically optimal structure for ORR, regardless of pH. However, the authors have only report the activities of Pt_xNi_y@Pt/C in acidic electrolyte, which are consistent with theoretical predictions. It remains unclear what the activities of Pt_xNi_y@Pt/C are in alkaline media and whether they follow the activity trend predicted by theory.

Response: Thank you very much for the kind suggestions. One of the major advancements of our quantitative microkinetic model lies in its universality in considering pH-dependent ORR performance. To bridge the gap between experimental results and theoretical predictions in alkaline media, we have evaluated ORR activity in a 0.1 M KOH electrolyte. The results, including CV curves, ORR polarization curves, Tafel plots, MA and SA, have been added to the **revised Supplementary Information** (Supplementary Fig. 11 and Supplementary Table 3). The above results confirm that the activity trend of Pt_xNi_y@Pt/C aligns with theoretical predictions, wherein the Pt₁Ni₁@Pt(111) exhibits the best ORR activity in alkaline media.

(3) The authors claim that the instability of com-Pt/C is associated with movement, aggregation, and Ostwald ripening process. Could this be the reason why commercial catalysts experience degradation after undergoing 10,000 cycles of stability testing? It is necessary to furnish supporting evidence for this claim.

Response: Thank you for the careful comments. The degradation mechanisms of Pt/C catalysts in acidic ORR have been previously reported to involve Ostwald ripening, agglomeration, carbon corrosion, particle dissolution, and detachment (Nat. Commun. 2022, 13, 7270; Energy Environ. Sci. 2023, 16, 1838-1869). Inspired by your suggestion, we further monitored the structural evolution of com-Pt/C after 30,000 ADT stability tests as described in the **revised Supplementary Information** (Supplementary Fig. 12). The TEM images reveal significant aggregation of Pt nanoparticles supported on carbon, resulting in the formation of larger particles with irregular morphologies and thus the observed deactivation of the com-Pt/C catalysts.

(4) Some minor errors: (1) In Fig. 3a, the PtNi@Pt/C should be Pt₁Ni₁@Pt/C? (2) The units of half-wave potential should be supplemented with the designation "vs. RHE". (3) The Supplementary Information contains a reference to AIMD results on line 9 of page 3, which incorrectly cites Fig. 4 in the manuscript instead of Fig. 5.

Response: Thank you very much for the careful comments. We have made the necessary corrections in the **revised Manuscript** (lines 4-5, page 10, and Fig. 3a and 4e) and **revised Supplementary Information** (line 9, page 3).

Response to Reviewer #2

Reviewer #2

In this paper, the authors present a synthesis method for PtNi/C catalysts with high dispersion and uniform size and propose a microkinetic model to predict the performance of these catalysts. The prepared catalysts exhibited very high activity in the oxygen reduction reaction (ORR) and demonstrated significantly superior durability compared to previously reported platinum alloy catalysts for the oxygen reduction reaction. However, the following aspects need to be addressed for the publication of this paper.

Response: We sincerely thank you for your positive evaluation on our work, especially the superior durability compared to previously reported platinum alloy catalysts for ORR. We also appreciate your constructive comments on guiding the revision of our manuscript. For your convenience, the main revisions are marked in red in the **revised Manuscript** and **Supplementary Information**.

(1) The synthesized PtNi/C catalysts underwent a dealloying process before the evaluation of ORR performance. Since dealloying removes nickel from the alloy particles, the actual platinum-to-nickel ratio of the catalysts is likely different. For the proposed microkinetic model to be appropriate, it is necessary to measure the platinum-to-nickel ratio of the dealloyed PtNi/C catalysts.

Response: Thank you very much for the valuable comments and suggestions. Yeah, we agree with you. Based on the quantitative microkinetic model, Pt₁Ni₁@Pt(111) core-shell catalyst is located at the theoretical maximum and shows the best ORR performance. The precise preparation of Pt₁Ni₁@Pt core-shell catalysts through appropriate synthetic methodologies is crucial in our study.

According to the TEM images and elemental mappings in Fig. 1c-e in the **revised Manuscript**, the as-prepared Pt₁Ni₁ alloy nanoparticles homogeneously deposit on carbon black with a consistent size and uniform distribution of Pt and Ni throughout the nanoparticle. We obtained Pt₁Ni₁@Pt core-shell catalysts with a Pt-rich shell and a Pt₁Ni₁ core through subsequent electrochemical dealloying process of as-prepared Pt₁Ni₁ alloy catalysts due to Ni leaching on the surface of the nanoparticles. Due to the surface Ni leaching and the formation of Pt-rich shell, the Pt-to-Ni ratio across the entire Pt₁Ni₁@Pt core-shell nanoparticle will inevitably increase after this dealloying process by comparing with that of as-prepared Pt₁Ni₁ nanoparticle. But the Pt-to-Ni ratio in the Pt₁Ni₁ core should remain close to 1:1.

To prove the above conclusion, the EDS result of dealloyed Pt₁Ni₁@Pt core-shell nanoparticle was added in the **revised Supplementary Information** (Supplementary Fig. 8). This result reveals a Pt-to-Ni ratio of 62.3:37.7 for the entire Pt₁Ni₁@Pt core-shell nanoparticle. However, this result does not accurately reflect the actual composition of the Pt₁Ni₁ core. Therefore, we roughly estimate the Pt-to-Ni ratio inside the Pt₁Ni₁ core through a simple calculation of a single nanoparticle:

We first construct a sphere model with radius (*r*) of ~1.5 nm, based on the average size of Pt₁Ni₁/C nanoparticles. The volume of a single particle (*V*_{particle}) is calculated as follows:

$$V_{particle} = \frac{4}{3}\pi r^3 = \frac{4}{3}\pi(1.5)^3 \approx 14.1 \text{ nm}^3$$

Then, the average atom volume (*V*_{atom}) can be estimated from unit cell volume, given that the volume of a face-centered cubic (fcc) unit cell is equal to the volume of four atoms:

$$V_{atom} = \frac{1}{4} a_{avg}^3$$

Where *a*_{avg} is the average lattice constant calculated from lattice constants of Pt (*a*_{Pt}) and Ni (*a*_{Ni}):

$$a_{avg} = \frac{1}{2}(a_{Pt} + a_{Ni}) = \frac{1}{2}(0.392 + 0.352) = 0.372 \text{ nm}$$

Therefore, the average atom volume (V_{atom}) is:

$$V_{atom} = \frac{1}{4}(0.372)^3 \approx 0.0129 \text{ nm}^3$$

The total number of atoms (N_{total}) in a single particle can be estimated as:

$$N_{total} = \frac{V_{particle}}{V_{atom}} \approx \frac{14.1}{0.0129} \approx 1093$$

Because the Pt-to-Ni ratio is ~1:1 before dealloying, thus, there are ~547 Pt atoms ($N_{Pt}=547$) and ~547 Ni atoms ($N_{Ni}=547$) in a single particle.

Assuming all the Ni atoms in the outermost monolayer dissolved after electrochemical dealloying while all the Pt atoms remain. Given that the thickness of the outermost monolayer can be estimated as one atomic layer, which is a_{avg} , and the volume of the outermost monolayer (V_{out}) can be calculated as follows:

$$V_{out} = \frac{4}{3}\pi \left((r)^3 - (r - a_{avg})^3 \right) \approx \frac{4}{3}\pi \left((1.5)^3 - (1.5 - 0.372)^3 \right) \approx 8.13 \text{ nm}^3$$

The dissolved number of Ni atoms ($N_{diss-Ni}$) in the outermost monolayer is:

$$N_{diss-Ni} = \frac{1}{2} \times \frac{V_{out}}{V_{avg}} \approx \frac{1}{2} \times \frac{8.13}{0.0129} \approx 315$$

The number of Ni ($N_{core-Ni}$) that remain in the Pt_1Ni_1 core is:

$$N_{core-Ni} = N_{Ni} - N_{diss-Ni} \approx 547 - 315 \approx 232$$

The Pt-to-Ni ratio ($ratio_{Pt-to-Ni}$) across the dealloyed $Pt_1Ni_1@Pt$ nanoparticle can be estimated as:

$$ratio_{Pt-to-Ni} = N_{Pt} : N_{core-Ni} \approx 547 : 232 \approx 70 : 30$$

It can be seen that in the sphere model, the Pt-to-Ni ratio in the Pt_1Ni_1 core remains 50:50, but the estimated Pt-to-Ni ratio across the entire particle increases to ~70:30. Notably, we assumed that all the Ni atoms in the outermost monolayer dissolved while the Pt atoms did not dissolve during the electrochemical dealloying process, so the Pt-to-Ni ratio of 70:30 is an overestimate. Indeed, the Pt-to-Ni ratio (62.3:37.7) obtained from EDS analysis of dealloyed $Pt_1Ni_1@Pt/C$ appears to be more reasonable. This analysis suggests that the Pt-to-Ni ratio in the Pt_1Ni_1 core remains 50:50, in line with theoretical predictions.

We have added the above discussion about Pt-to-Ni ratio of $Pt_1Ni_1@Pt$ core-shell nanoparticle in the **revised Manuscript** (lines 3-6, page 9) and **revised Supplementary Information** (Supplementary Notes, pages 3-4).

(2) The crystallographic characteristics of the prepared catalysts need to be provided. It is essential to measure and discuss the XRD of the catalysts before and after dealloying.

Response: Thank you very much for the important comment. In the **revised Supplementary Information** (Supplementary Fig. 9), we have included XRD patterns of Pt_1Ni_1/C NPs before and after the dealloying process. These patterns reveal a diffraction peak corresponding to the (111) facet, which is situated between pure Pt (111) and pure Ni (111). The widening of the diffraction peak can be attributed to the ultra-small size of Pt_1Ni_1 NPs. The aberration-corrected HADDF-STEM image shows as well as (111) facet corresponding to face-centered cube (fcc) structures, consistent with the XRD results.

(3) It is known that the durability degradation of platinum alloy catalysts is more influenced by the leaching of non-precious metal components than by the aggregation of alloy particles. However, in this paper, the authors claim that the high durability of the synthesized PtNi/C catalysts is due to the suppression of aggregation by the strong interaction between carbon (defect sites) and platinum alloy particles. This needs to be clarified through a compositional analysis of the catalysts after the durability tests.

Response: Thank you very much for your valuable comment. We appreciate your constructive opinion on the degradation mechanism of Pt alloy catalysts. Yes, the durability of the ORR is easily affected by the extensive leaching of non-noble metal components. In our study, we found that the effective suppression of aggregation of the Pt₁Ni₁@Pt nanoparticles also play an important role in ORR stability.

To track the composition evolution of the catalysts after the durability tests, we examined the composition variation of the catalysts after 10,000, 30,000, 50,000, and 70,000 ADT cycles, respectively. The results are now included in the **revised Supplementary Information** (Supplementary Fig. 15), revealing a gradual decrease in Ni species during the ADT test. The ongoing etching of Ni has resulted in significant compositional alterations in atomic ratio of Ni from 37.7% to 8.6% during ADT tests. However, the changes in composition do not adversely affect the activity after 70,000 ADT cycles. So, we concluded that the superior structural stability of the synthesized Pt₁Ni₁@Pt/C catalysts is crucial for the stable ORR, which may compensate for the leaching of non-precious metal components.

(4) If the defect sites generated during the catalyst preparation inhibit metal aggregation during durability tests, then the synthesized Pt/C should exhibit similar durability behavior as Pt₁Ni₁/C. Therefore, the authors should conduct durability tests on the synthesized Pt/C catalysts and provide an examination of the morphology before and after the tests.

Response: Thank you very much for your helpful suggestion. We conducted durability tests on syn-Pt/C catalysts and compared their morphology changes before and after the durability tests. The results demonstrated an enhanced durability of syn-Pt/C similar to that of Pt₁Ni₁@Pt/C. The TEM images of syn-Pt/C catalysts before and after durability test are provided in the **revised Supplementary Information** (Supplementary Fig. 14), demonstrating minimal particle aggregation or movement. These findings validate the improved durability of syn-Pt/C, which can be mainly attributed to the strong Pt-C interaction.

(5) In the durability tests, the authors used a Pt foil as the counter electrode. Using a Pt foil as the counter electrode, especially in durability tests, makes accurate evaluation difficult due to platinum redeposition. Therefore, authors should discuss the result of the durability test conducted with a carbon as counter electrode.

Response: Thank you very much for your kind suggestion. Yes, the utilization of Pt foil as the counter electrode can give rise to the issue of Pt redeposition. It has been well-documented that using a Pt counter electrode can lead to problematic ORR performance for non-precious metal-based catalysts (Chin. J. Catal., 2016, 37, 1109-1118; Appl. Catal., A 2019, 588, 117273), due to Pt species dissolving from the counter electrode and redepositing on the working electrode. So, a carbon counter electrode is often employed to avoid this issue for non-precious metal-based catalysts (Nat Catal. 2024, 7, 139-147; Nat Commun. 2018, 9, 5422). However, for Pt-based catalysts, this

approach will introduce another challenge that the carbon monoxide poisoning resulting from oxidation of the carbon electrode adversely affect the ORR performance (ACS Catal. 2020, 10, 10773-10783).

The ORR activity of Pt₁Ni₁@Pt is further investigated using a carbon counter electrode. As shown in Fig. R2, the ORR polarization curves illustrate that the activity of Pt₁Ni₁@Pt rapidly decreases after only 50 or 150 ADT cycles.

Fig. R2 ORR polarization curves of Pt₁Ni₁@Pt using carbon rod as a counter electrode.

Considering the benchmarking evaluation of ORR performance for Pt-based catalysts on RDE setups and the utilization of Pt counter electrode in most reported studies focusing on noble-metal based catalysts (Science 2021, 374, 459-464; Nat. Nanotechnol. 2022, 17, 968-975; Electrochim. Acta 2015, 179, 647-657), the influence of Pt on ORR performance appears to be relatively insignificant. So, the use of Pt foil as the counter electrode may be more advantageous in our experimental setup.

(6) Catalysts that exhibit high performance in half cell do not always perform similarly in unit cells. Since the platinum alloy catalysts in this paper are intended for fuel cells, the performance evaluation results of unit cells using the synthesized catalysts should be provided.

Response: Thank you very much for your kind suggestion. We agree that the catalytic performance in a half-cell does not always reflect the practical potential in fuel cells. Performance evaluation in a PEMFC is indeed necessary and beneficial. The detailed conditions for evaluating PEMFC performance are as follows:

The cathode catalyst (Pt₁Ni₁/C) was mixed with Nafion D520 ionomer at an ionomer-to-carbon ratio of 0.3 in isopropanol and ultrasonicated for 1 hour to form a homogeneous ink. The catalyst-coated-membrane (CCM) with an active area of 5 cm² was prepared on Gore membrane (M735.18) using a spray-coating method. The anode catalyst (commercial Pt/C, Johnson Matthey, 40 wt.%) was also sprayed onto the other side of the membrane using a similar method. The Pt loadings were 0.2 mg_{Pt} cm⁻² at the anode and 0.1 mg_{Pt} cm⁻² at the cathode. A gas diffusion layer (YLS-30T) was pressed with the CCM and gaskets to obtain the membrane electrode assembly.

Fuel cell testing was performed in a single cell using a commercial fuel cell test system (Scribner 850e). The break-in process involved cycling between 0.85 V and 0.4 V under H₂-air condition until

a stable current was obtained. The polarization curves were collected at 80°C, 100% relative humidity (RH), with a backpressure of 250 kPa_{abs} and a gas flow rate of 500 sccm H₂/2000 sccm air for the anode/cathode. As shown in **Fig. R3**, the Pt₁Ni₁/C catalysts show a highest peak power density of 1.32 W cm⁻².

Fig. R3 H₂/air PEMFC performance of Pt₁Ni₁/C catalysts.

Reviewer #3 (Remarks to the Author)

Cui et al. used a quantitative microkinetic model to estimate the oxygen reduction reaction (ORR) performance of $\text{Pt}_x\text{Ni}_y@\text{Pt}(111)$ and synthesized the corresponding catalysts using a sonochemical approach. This work lacks novelty, as similar results from simulations and experiments have been reported frequently in the literature. The highlighted importance of Pt-C interactions in enhancing ORR catalyst durability is also validated through fuel cell tests. I recommend rejection of this manuscript.

Response: We appreciate your comments and suggestions on guiding the revision of our manuscript. Regarding the theoretical modeling aspect, we would like to emphasize that our work presents significant innovation. Compared to traditional thermodynamic (J. Phy. Chem. B, 2004, 108, 17886-17892) or kinetic volcano plots (J. Phy. Chem. C, 2014, 118, 6706-6718), our model also considers the influence of pH. To the best of our knowledge, this is the first time a pH-dependent volcano plot for Pt-based catalysts has been used to guide the design of Pt-based alloy catalysts. This represents a notable advancement over previous theoretical models. Moreover, the challenge of accurately synthesizing pre-designed structures that closely resemble the theoretical prediction usually poses another significant obstacle in practical implementation. Here, we developed a fast (within a few minutes), high-intensity sonochemical synthesis method based on rapid heating rates and thus localized high temperature. The acoustic cavitation effect from sonochemical treatment ensures the successful formation of ultra-small Pt_xNi_y alloy NPs and highly defective graphite, thereby facilitating a strong interaction with Pt atoms. Therefore, owing to its exceptional structural stability, the $\text{Pt}_1\text{Ni}_1@\text{Pt}/\text{C}$ exhibits a superior ORR durability with negligible degradation in activity (only 1.6%) over 70000 potential cycles, without any discernible migration or aggregation of the $\text{Pt}_1\text{Ni}_1@\text{Pt}$ NPs. For your convenience, the main revisions are marked in red in the **revised Manuscript** and **Supplementary Information**.

1. Why the simulated polarization curves for $\text{Pt}_3\text{Ni}(111)$ and $\text{Pt}_1\text{Ni}_1(111)$ in Fig.1b are not parallel.

Response: Thank you very much for your careful comment. The slight difference in the polarization curves for $\text{Pt}_3\text{Ni}(111)$ and $\text{Pt}_1\text{Ni}_1(111)$ in Fig. 1b in the **revised Manuscript**, which are not perfectly parallel, can be explained as follows:

As seen in Fig. 1a in the **revised Manuscript**, the descriptor G_{HO^*} for $\text{Pt}_3\text{Ni}(111)$ is slightly larger than that of $\text{Pt}_1\text{Ni}_1(111)$. Based on the linear scaling relationship ($G_{\text{O}_2^*} = 1.2G_{\text{HO}^*} + 4.09$) (J. Phy. Chem. C, 2020, 124,14581-14591), we can infer that O_2^* adsorption on $\text{Pt}_3\text{Ni}(111)$ would be somewhat weaker, as shown in **Fig. R4**. Consequently, at the same electrode potential, the reaction rate of the $\text{O}_2 \rightarrow \text{HO}_2^*$ step on the $\text{Pt}_3\text{Ni}(111)$ surface would be slower. This ultimately results in the subtle difference between these two curves.

Fig R4. Free energy diagram of Pt₃Ni₁ and Pt₁Ni₁ at 0.9 V/RHE.

2. There seems to be contamination affecting the durability test of Pt/C (Fig.6d). The observed degradation behavior is unusual and merits further investigation.

Response: Thank you very much for the kind suggestion. Yes, the polarization curves for com-Pt/C were indeed incorrectly plotted, possibly due to contamination in the electrolyte or the glassware. This resulted in the unusual shape of the polarization curves. The durability test data for com-Pt/C in Fig. 4c-4e has been updated in the **revised Manuscript**. Consequently, we have made corrections regarding the observed performance degradation of com-Pt/C in the **revised Manuscript**.

3. The authors should include X-ray diffraction (XRD) results for the Pt_xNi_y@Pt(111) catalysts to provide more comprehensive characterization data.

Response: Thank you very much for your suggestion. We have added the XRD patterns of Pt₁Ni₁/C catalysts before and after electrochemical dealloying process in the revised **Supplementary Information** (Supplementary Fig. 9), which shows a diffraction peak corresponding to (111) facet located between Pt(111) and Ni(111). Notably, the diffraction peak widens due to the ultra-small size of Pt₁Ni₁ and Pt₁Ni₁@Pt NPs. The aberration-corrected HADDF-STEM image shows as well as (111) facet corresponding to face-centered cube (fcc) structures, consistent with the XRD results.

4. The authors attribute the partial embedding of Pt particles within the carbon matrix to strong Pt-C bonds (as indicated in Fig. 5f). This explanation seems inaccurate; partial embedding is often due to overlapping and is commonly observed in commercial Pt/C catalysts.

Response: Thank you very much for the important comments. Yes, the phenomenon of partial embedding of particles within carbon supports has been previously reported, typically attributed to carbon matrix transformation induced by transient high temperatures such as thermal shock or Joule heating method (Nano Lett. 2016, 16, 5553-5558; Nat. Commun. 2020, 11, 6373). Considering the transient high temperatures associated with our sonochemical synthesis method, resembling the thermal shock or Joule heating approach, we initially suspected an analogous phenomenon in our work. So, to mitigate potential misinterpretation of Pt particle overlapping with carbon as partial embedding, we carefully conducted a series of characterizations.

The formation of strong Pt-C bonds has been extensively demonstrated by XAS, XPS, and AIMD simulations. Particularly, the FT-EXAFS spectra at Pt L₃-edge (Fig. 5b in the **revised Manuscript**) displays the first Pt-C coordination peak in Pt₁Ni₁/C. Therefore, the observed partial embedding of Pt particles within the carbon matrix in Fig. 5f in the **revised Manuscript** serves as additional evidence reinforcing the presence of strong bonding between Pt and the carbon support. In addition, TEM studies conducted on Pt₁Ni₁@Pt/C and syn-Pt/C catalysts before and after ADT tests demonstrated negligible alterations in the morphology and size (Supplementary Figs. 13-14 in the **revised Supplementary Information**). However, the com-Pt/C catalyst exhibited a tendency for Pt NPs to aggregate or migrate (Supplementary Fig. 12 in the **revised Supplementary Information**) after ADT tests, possibly due to the overlapping of Pt NPs with the carbon support. Thus, we concluded that the Pt₁Ni₁@Pt/C and syn-Pt/C catalysts exhibit a strong Pt-C interaction, effectively inhibiting the migration or aggregation of the NPs even after undergoing stability tests over 70000 potential cycles, unlike the overlapping observed in com-Pt/C catalysts.

REVIEWER COMMENTS

Reviewer #1 (Remarks to the Author):

The authors have offered a comprehensive and considerate response to my comments, which should significantly improve the quality and acceptance potential of their manuscript after reasonably addressing the comments of other reviewers.

Reviewer #2 (Remarks to the Author):

The authors have conducted supplementary experiments to address all issues raised by the reviewer and have appropriately revised and improved the manuscript.

Reviewer #3 (Remarks to the Author):

The authors have addressed most of questions. I think the authors should include the fuel cell test result in the manuscript. And It will be better if they can validate the durability in fuel cell level. Currently, liquid cell result is hard to demonstrate the performance of catalyst.

RESPONSE TO REVIEWERS

Reviewer #1

The authors have offered a comprehensive and considerate response to my comments, which should significantly improve the quality and acceptance potential of their manuscript after reasonably addressing the comments of other reviewers.

Reponse: Thank you very much for your approval of our work.

Reviewer #2

The authors have conducted supplementary experiments to address all issues raised by the reviewer and have appropriately revised and improved the manuscript.

Reponse: Thank you very much for your approval of our work.

Reviewer #3

The authors have addressed most of questions. I think the authors should include the fuel cell test result in the manuscript. And it will be better if they can validate the durability in fuel cell level. Currently, liquid cell result is hard to demonstrate the performance of catalyst.

Response: Thank you very much for your important and valuable suggestions. Yes, the discrepancies between results obtained in liquid cells and that in fuel cell environments are existed. It is important to validate the performance and durability of the catalyst at the fuel cell level.

During the last revision, we conducted a PEMFC test to response to the comments by the second reviewer, but, we did not assess its stability. With the generous assistance of Prof. Gao at the University of Science and Technology of China, we were able to conduct our first PEMFC test during the last revision. The detailed conditions for evaluating PEMFC performance are as follows: the cathode catalyst (Pt₁Ni₁/C) was mixed with Nafion D520 ionomer at an ionomer-to-carbon ratio of 0.3 in isopropanol and ultrasonicated for 1 hour to form a homogeneous ink. The catalyst-coated-membrane (CCM) with an active area of 5 cm² was prepared on Gore membrane (M735.18) using a spray-coating method. The anode catalyst (commercial Pt/C, Johnson Matthey, 40 wt.%) was also sprayed onto the other side of the membrane using a similar method. The Pt loadings were 0.2 mg_{Pt} cm⁻² at the anode and 0.1 mg_{Pt} cm⁻² at the cathode. A gas diffusion layer (YLS-30T) was pressed with the CCM and gaskets to obtain the membrane electrode assembly.

Fuel cell testing was performed in a single cell using a commercial fuel cell test system (Scribner 850e). The break-in process involved cycling between 0.85 V and 0.4 V under H₂-air condition until a stable current was obtained. The polarization curves were collected at 80°C, 100% relative humidity (RH), with a backpressure of 250 kPa_{abs} and a gas flow rate of 500 sccm H₂/2000 sccm air for the anode/cathode. As shown in **Fig. R1**, the Pt₁Ni₁/C catalysts show a highest peak power density of 1.32 W cm⁻².

Fig. R1 H₂/air PEMFC performance of Pt₁Ni₁/C catalysts.

Unfortunately, the instruments in Prof. Gao lab recently encountered technical malfunctions, therefore, we sought assistance from an authorized third-party fuel cell testing company on this occasion to have the second PEMFC tests and assess its stability. The experimental details are as follows: the cathode catalyst (Pt₁Ni₁@Pt/C) was mixed with Nafion D520 ionomer at an ionomer-to-carbon ratio of 0.3 in isopropanol and ultrasonicated for 1 hour to form a homogeneous ink. The CCM with an active area of 4 cm² was prepared on a proton exchange membrane (Gore M735.18, thickness: 18 μm) using a spray-coating method. The anode catalyst (commercial Pt/C, Johnson Matthey, 40 wt.%) was similarly sprayed onto the other side of the membrane. The Pt loadings were 0.2 mg_{Pt} cm⁻² at the anode and 0.1 mg_{Pt} cm⁻² at the cathode. Two gas diffusion layers (H15CX483 for the cathode and H14CX653 for the anode, both from Freudenberg) was pressed with the CCM and gaskets to obtain the membrane electrode assembly (MEA).

Fuel cell tests were conducted in a single cell with 13-channel straight flow field using a commercial fuel cell test system (Greenlight G40). The break-in process involved cycling between 0.75 V and 0.3 V under H₂-air condition (0.5 NLPM/2 NLPM) with a back pressure of 250 kPa_{abs} until a stable current was obtained. The polarization curves were collected at 80°C, 100% relative humidity (RH), with a back pressure of 250 kPa_{abs} and a gas flow rate of 0.5 NLPM H₂/2 NLPM air for the anode/cathode. Accelerated durability test (ADT) was performed by square-wave voltammetry between 0.60 and 0.95 V for up to 30,000 cycles (holding 2.5 s for each voltage level, H₂/N₂ at 0.2/0.2 NLPM, 101 kPa, 80 ° C, 100% RH), following US DOE MEA ADT protocol for PGM-based catalysts.

The Pt₁Ni₁/C catalysts tested in the fuel cell test system (Greenlight G40) exhibit a peak power density of 1.08 W cm⁻², as illustrated in Fig. R2. However, this value is lower compared to that obtained from the fuel cell test system (Scribner 850e). Such differences suggests that the performance of PEMFC is not solely determined by the catalyst itself but also influenced by the PEMFC test instruments, MEA fabrication and PEMFC test processes, including, including PEM selection, CCM fabrication, ionomer/carbon (I/C) ratio, catalyst ink formulation, GDL selection,

compression pressure, flow field design, and break-in process (Sci. Adv. 2017, 3 (10), eaa0476; Chem. Soc. Rev. 2022, 51 (4), 1529-1546; Int. J. Hydrogen Energy, 2010, 35 (5), 2119-2126; Int. J. Hydrogen Energy, 2010, 35 (17), 9349-9384.). The achievement of a stable and outstanding result in PEMFC testing often requires a wealth of experience and significant trial-and-error efforts for optimization, most of which typically take at least half of one year or even longer.

Regarding durability, we were only able to compare the polarization curves before and after ADT, as we lack the capability and expertise to rigorously measure mass activity or rated power density. The peak power density of Pt₁Ni₁@Pt/C showed a degradation of approximately 42% at 250 kPa_{abs} (Fig. R2). Durability at the fuel cell level can be influenced by numerous factors, including membrane degradation, Pt dissolution, particle agglomeration, carbon support corrosion, contaminant ingress, flooding or drying, GDL degradation, and bipolar plate degradation, etc. (J. Electrochem. Soc., 2018, 165 (6), F3148-F3160; Curr. Opin. Electrochem., 2020, 21, 192-200; Int. J. Hydrogen Energy 2021, 46 (22), 12206-12229). Given that the PEMFC performance was not fully optimized and the aforementioned factors were not thoroughly controlled, the observed performance degradation may not accurately reflect the intrinsic stability of Pt₁Ni₁@Pt/C.

Fig. R2. H₂/air PEMFC performance of Pt₁Ni₁@Pt/C before and after ADT test at 250 kPa_{abs}.

We must acknowledge that our team currently lacks the requisite experience in conducting ORR performance evaluations at the fuel cell level, particularly in areas such as MEA fabrication, including CCM preparation, hot pressing, and sealing. Additionally, we do not have access to a fuel cell testing system, nor do we possess sufficient operational experience to accurately assess PEMFC performance. In this situation, our catalyst has shown good PEMFC performance (a peak power density of 1.08 W cm⁻² at the fuel cell test system (Greenlight G40) and 1.32 W cm⁻² at the fuel cell test system (Scribner 850e)). We believe that the performance of Pt₁Ni₁@Pt/C obtained in this study may have been underestimated and holds significant potential for further optimization.

After conducting preliminary PEMFC tests, we can confirm the material's good activity in PEMFC and its significant potential for optimization in the PEMFC performance and especially the durability. However, this dataset is not optimal, rendering the inclusion of such immature data in the

manuscript inappropriate. Nonetheless, inspired by the reviewer's suggestions, we have actually initiated equipment procurement and plan to dedicate to PEMFC testing.

In addition, we believe that the existing stability measurements performed using RDE in a liquid cell provide substantial support for our key finding regarding the strong Pt-C bonds that inhibits nanoparticle aggregation. While validating durability at the fuel cell level is certainly important, we believe it is not indispensable for the key conclusions of our study. We regret that we are unable to include publishable fuel cell level experimental results in this revised manuscript, but we are actively working to address this limitation. We anticipate being able to perform these tests in the near future once our purchased commercial fuel cell testing system is operational.

REVIEWERS' COMMENTS

Reviewer #3 (Remarks to the Author):

This manuscript can be as it is.